# Small Stochastic Data Compactification Concept Justified in the Entropy Basis

**DOI:** 10.3390/e25121567

**Published:** 2023-11-21

**Authors:** Viacheslav Kovtun, Elena Zaitseva, Vitaly Levashenko, Krzysztof Grochla, Oksana Kovtun

**Affiliations:** 1Internet of Things Group, Institute of Theoretical and Applied Informatics Polish Academy of Sciences, Bałtycka 5, 44-100 Gliwice, Poland; 2Department of Informatics, University of Žilina, 010 26 Žilina, Slovakia; 3Department of the Theory and Practice of Translation, Faculty of Foreign Languages, Vasyl’ Stus Donetsk National University, 600-Richchya Str., 21, 21000 Vinnytsia, Ukraine

**Keywords:** machine learning, data analysis, entropy, data reliability, small data, stochastic data, compactification, completeness, parametric optimization

## Abstract

Measurement is a typical way of gathering information about an investigated object, generalized by a finite set of characteristic parameters. The result of each iteration of the measurement is an instance of the class of the investigated object in the form of a set of values of characteristic parameters. An ordered set of instances forms a collection whose dimensionality for a real object is a factor that cannot be ignored. Managing the dimensionality of data collections, as well as classification, regression, and clustering, are fundamental problems for machine learning. Compactification is the approximation of the original data collection by an equivalent collection (with a reduced dimension of characteristic parameters) with the control of accompanying information capacity losses. Related to compactification is the data completeness verifying procedure, which is characteristic of the data reliability assessment. If there are stochastic parameters among the initial data collection characteristic parameters, the compactification procedure becomes more complicated. To take this into account, this study proposes a model of a structured collection of stochastic data defined in terms of relative entropy. The compactification of such a data model is formalized by an iterative procedure aimed at maximizing the relative entropy of sequential implementation of direct and reverse projections of data collections, taking into account the estimates of the probability distribution densities of their attributes. The procedure for approximating the relative entropy function of compactification to reduce the computational complexity of the latter is proposed. To qualitatively assess compactification this study undertakes a formal analysis that uses data collection information capacity and the absolute and relative share of information losses due to compaction as its metrics. Taking into account the semantic connection of compactification and completeness, the proposed metric is also relevant for the task of assessing data reliability. Testing the proposed compactification procedure proved both its stability and efficiency in comparison with previously used analogues, such as the principal component analysis method and the random projection method.

## 1. Introduction

The most valuable resource in the information society is data. It seems that “there is no such thing as too much data”, but let us try to look at this catchphrase as data scientists. The “curse of dimensionality” is a problem that consists of the exponential growth of the amount of data that has occurred simultaneously with the growth of the dimensionality of the space for data representation. This term was introduced by Richard Bellman in 1961. Scientists dealing with mathematical modelling and computational methods were the first to face this problem. Now, this problem is faced again as machine learning and artificial intelligence methods are implemented. In this study, we will illustrate the relevance of this problem using the k-nearest neighbour method, which is popular for solving classification problems [1,2,3,4]. The essence of the method is as follows: the instance belongs to the same class as that which the majority of its nearest neighbour instances in the parametric space belong. To ensure high-quality work with this method, the saturation density of the parametric space with instances must be sufficiently high. How are the parametric space dimensions, the density of instances, and their number related to each other? To uniformly cover a unit interval [0, 1] with a density 0,01, we need 100 points, where the coverage density is defined as the ratio of the number of points evenly distributed in the target interval to the length of the latter. Now, imagine a 10-dimensional cube. To achieve the same coverage density, we already need 10^20^ points, that is, 10^18^ times more points compared to the original 1-dimensional space. This example demonstrates the reason for the inefficiency of the brute force method in typical machine learning problems (classification, clustering, and regression) [5,6,7,8,9]. The paradox is that it is impossible to solve the mentioned applied problems using a small number of parameters and achieve adequate results. One can simply turn a blind eye to the problem of dimensionality, which is the paradigm of deep learning, where using non-parameterized models achieves a significant increase in their quality despite the colossal increase in the number of calculations and accepting as an axiom the potential instability of the training process. But this recipe is unacceptable in the context of the machine learning ideology. The following Table 1 contains a more detailed comparison of these two methods.

Therefore, managing the dimensionality of data while preserving their quality and the representativeness of the parametric space is an urgent scientific problem for machine learning.

The most widely used method for reducing data dimensionality is singular value decomposition (SVD, [10,11,12]). The matrices obtained as a result of SVD have a very specific interpretation in the machine learning methodology. They can be used according to the proven method both for principal component analysis (PCA, [13,14,15]) and (with certain reservations) for non-negative matrix factorization (NMF, [16,17,18]). SVD can also be used to improve the results of independent component analysis (ICA, [19,20,21]). It is convenient to apply SVD because there are no restrictions on the structure of the original data matrix (square when using the LU [22] or Schur distribution [23]; square, symmetric, or positive definite when using the Cholesky distribution [24]; matrix with positive elements when applying NMF). The essence of SVD is the representation of the original matrix *X* as a product of matrices of the form $X=U\Sigma V^{*}$, where *U* is a unitary matrix of order *m* and ∑ is a rectangular diagonal matrix of dimension (*m* × *n*), where *m* is a number of instances and *n* is a number of measured observables, with singular elements on the main diagonal and $V^{*}$ is a matrix of order *n*, obtained as a result of conjugate transpose of the matrix *V*. The matrix ∑ is important for the dimensionality management problem. The squared singular elements of this matrix are interpreted as the variance *σ*^2^ of the corresponding component. Based on the value of these variances, the researcher can select the required number of components. What is the best value $\sum_{m}\sigma^{2}$? Some recommend maintaining the inequality $\sum_{m}\sigma^{2}\geq 0,90$, while others believe that $\sum_{m}\sigma^{2}\geq 0,50$ is sufficient. The original answer to this question is provided by Horn’s parallel analysis based on Monte Carlo simulation [25]. The disadvantage of both SVD and PCA is the high computational complexity of obtaining a singular distribution (well-known randomized algorithms [26] slightly mitigate this limitation). A more serious limitation is the sensitivity of SVD/PCA to outliers and the type of distribution of the original data. Most researchers believe that SVD/PCA works consistently with normally distributed data, but it has been empirically found that, as the data dimensionality increases, there are exceptions even to this rule. Therefore, SVD/PCA methods cannot guarantee the stability of the data dimensionality reduction procedure.

NMF is used to obtain the decomposition of a non-negative matrix $X_{\left(m\times n\right)}$ into non-negative matrices $W_{\left(m\times k\right)}$ and $H_{\left(k\times n\right)}$: *X* = *WH*. By choosing k<<m,n, we can solve the problem of reducing the dimensionality of the original matrix quite effectively. The problem is that, unlike SVD, finding the *X* = *WH* decomposition does not have an exact solution. There are specialized formulations of quadratic programming problems, such as the support vector machine (SVM, [27,28,29]) [30]. However, we understand that this means that NMF has the same limitations that have been pointed out for SVD/PCA.

The ICA method crossed into machine learning from the signal processing theory and, in its original formulation, was intended for the decomposition of a signal with additive components. At the same time, it was believed that these components have an abnormal distribution, and the sources of their origin are independent. To determine independent components, either minimization of mutual information based on Kullback–Leibler divergence [19] or minimization of “non-Gaussianity” [20,21] (using measures such as kurtosis coefficient and negentropy) are used. In the context of the dimensionality reduction problem, the application of ICA is trivial: to represent the input data as a mixture of components, divide them and select a certain number. There is no analytically consistent criterion for component selection.

We have often mentioned machine learning methods in the context of the data dimensionality management problem. However, there are competitors originating from the artificial intelligence field, i.e., the autoencoders [31,32,33]. This is an original class of neural networks, created so that the signal given to the input layer is reproduced as accurately as possible at the output of the neural network. The number of hidden layers should be at least one, and the activation functions of neurons on these layers should be non-linear (most often *sigmoid*, *tanh*, *ReLu*). If the number of neurons in the hidden layer is less than the number of neurons in the input layer, and we reproduce the input signal at the same time with sufficient accuracy as the output of the trained autoencoder, then the parameters of the neurons of the hidden layer are a compact approximate representation of the input signal. The advantage of this approach is that the neural network works for us. It is also very easy to orient the autoencoder to solve the data dimensionality increasing problem: it is sufficient that there are more neurons on the hidden layer than on the input layer. Disadvantages are also known: empirical search for the optimal configuration of the neural network (number of hidden layers, number of neurons on those layers, and selection of their activation functions), empirical selection of both the training algorithm and its parameters), and the neural network regularization methods (*L1*, *L2*, *dropout*). And we have not yet focused on the specific drawback of autoencoders, i.e., the tendency to degenerate hidden layers in the training process.

In recent years, there has been a growing interest in the research of data analysis, particularly within the context of regression analysis applied to inhomogeneous datasets. The existing research [34] explores the challenges presented by data that can be gathered from various sources or recorded at different time intervals, resulting in inherent inhomogeneities that complicate the process of regression modelling. The conventional framework of independent and identically distributed errors, typically associated with a single underlying model, is inadequate for handling such data. As the authors claim, traditional alternatives, like time-varying coefficients models or mixture models, can be computationally burdensome and impractical. So, the paper [34] proposes an aggregation technique based on normalized entropy (neagging) in contrast with such well-known aggregation procedures as bagging and magging. This approach has shown great promise, and the paper provides practical examples to illustrate its effectiveness using real-world datasets across various scenarios. However, the authors position their solution for working with large amounts of data or Big data. The issue of applicability of the mentioned procedures for compactification of small variable data has not been considered.

Taking into account the strengths and weaknesses of the mentioned methods, we will formulate the necessary attributes of scientific research.

The research object is the process of stochastic empirical data collection compactification.

The research subjects are probability theory and mathematical statistics, information theory, computational methods, mathematical programming methods, and experiment planning theory.

The research purpose is to formalize the process of finding the optimal probability distribution density of stochastic characteristic parameters of the empirical data compactification model with the maximum relative entropy between the original and compactified entities.

The research objectives are:-formalize the concept of calculating the variable entropy estimation of the probability distribution density of the characteristic parameters of the stochastic empirical data collection;-formalize the process of the stochastic empirical data collection compactification with the maximization of the relative entropy between the original and compactified entities;-justify the adequacy of the proposed mathematical apparatus and demonstrate its functionality with an example.

The Motivation. One derives quantitative information on a class of objects by measuring a set of observables (“characteristic parameters”) on a sample of objects taken from the class of interest. A set of values taken by the chosen observables on one of the objects is an instance. One of the basic problems in general data analysis is finding the optimal number of instances and the optimal (minimal) number of observables, that allow, in the presence of noise, to build regression models, estimate correlations between observables, and classify and cluster the objects in a machine learning approach. In this perspective, which is a very relevant one, the authors propose a model of noisy data based on a conditional, relative entropy [Equation (6)]. The article introduces a consistent and tunable method of “compactification” that performs quite well concerning other established methods, such as PCA and random projection methods.

## 2. Models and Methods

### 2.1. Statement of the Research

Let us characterize the researched process using a model in terms of linear programming, that is, by a function z=fv,w that summarizes *n* weighted characteristic parameters v∈ℝn, where the weights *w* are interval stochastic values: w∈W=w−≤w≤w+, the properties of which are characterized by the probability distribution density *P*(*w*).

Suppose that, as a result of *m* observations of the investigated process, empirical data with the structure V,y were obtained, where *V* is the training collection and each empirical parametric vector vi=vi1,…,vin∈V, vi∈ℝn, corresponds to an empirical initial value yi∈y, ∀i=1,m¯. When substituting data *V* into the model *z*, the equality of
(1)z=zi=Vw, i=1,m¯,
must be fulfilled and which is provided by the training of the model *z*.

We consider that the values *y_i_* of the original empirical vector *y* contain interference, which are represented by stochastic vector values $\varepsilon_{i}\in \varepsilon$, $i=\bar{1,m}$, $\varepsilon \in E=\left\{\varepsilon^{-}\leq \varepsilon \leq \varepsilon^{+}\right\}$, with the probability density function $L\left(\varepsilon \right)$ of a stochastic vector ε. Taking into account interferences, we present expression (1) as
(2)$$u=z+\varepsilon =V_{\left(m\times n\right)}w+\varepsilon ,$$
where $u\in U=\left[u^{-},u^{+}\right]$, $u^{-}=Vw^{-}+\varepsilon^{-}$, $u^{+}=Vw^{+}+\varepsilon^{+}$.

In the context of the formulated equation, the machine learning methodology is focused on determining the estimates $\overset{⌢}{P}\left(w\right)$ and $\overset{⌢}{L}\left(\varepsilon \right)$ of the corresponding probability distribution densities. The basis for this is model (2) and a set of empirical data *V*. Based on the known estimates of $\overset{⌢}{P}\left(w\right)$ and $\overset{⌢}{L}\left(\varepsilon \right)$, it is possible to outline the domain of stochastic vectors u∈U. Such a problem will be referred to as a *d*-problem. The authors devoted the article [35] directly to the solution of the *d*-problem.

On the other hand, the problem of compactification of the parametric space *V* of model (2) is solved by reducing the dimension of the characteristic parameters from *n* to *r* units, *r* < *n*, is also of practical value. Such a problem will be referred to as a *c*-problem.

Suppose that, as a result of the compactification of the original empirical data with the structure V,y, a shortened parametric space ℝr is obtained where each parametric vector yi=vi1,…,vir∈Y, yi∈ℝr, or i=1,m¯, corresponds to the original interval stochastic value a∈A=a−≤a≤a+, j=1,r¯, with the probability distribution density *A*(*a*).

To describe compactified data $\langle Y,a\rangle$, we define the model
(3)$$b=Y_{\left(m\times r\right)}a,a\in R^{r},b\in R^{m},$$
and the vector of observations is expressed as
(4)s=b+ξ,
where the stochastic vector ξ is formed by interval values $\Xi =\left\{\xi^{-}\leq \xi \leq \xi^{+}\right\}$ with the probability distribution density $Z\left(\xi \right)$. The vectors *s* defined by expression (4) are interpreted as $S=\left[s^{-},s^{+}\right]$, $s^{-}=Ya^{-}+\xi^{-}$, $s^{+}=Ya^{+}+\xi^{+}$.

Our further actions will be aimed at formulating:-optimality criterion of the compactified data matrix *Y*_(*m*×*r*)_;-a method for calculating the elements of the optimal compactified data matrix *Y*_(*m*×*r*)_;-a method for comparing the probability distribution densities of outputs of models (2) and (4) as an indicator of the effectiveness of the proposed compactification concept.

### 2.2. The Concept of Entropy-Optimal Compactification of Stochastic Empirical Data

Let us focus on the analytical formalization of the entropic properties of empirical data, summarized by the matrix *V*. Let there be *m* independent instances in the collection of class *X*, each of which is characterized by the values of *n* attributes (characteristic parameters). The selection of instances in the collection *X* is random. In this context, the matrix *X* summarizes $x_{ij}$, $i=\bar{1,m}$, $j=\bar{1,n}$, stochastic attributes whose values are real numbers: $x_{ij}\geq 0$, $i=\bar{1,m}$, $j=\bar{1,n}$, satisfying the condition $\sum_{i=1}^{m}\sum_{j=1}^{n}x_{ij}\leq W$, where *W* is determined by the region of origin of instances of the class *X*.

We normalize the values of the elements of the matrix X relative to the selected scale with a resolution of Δ: $h_{ij}=\left\lceil x_{ij}/\Delta \right\rceil$, $i=\bar{1,m}$, $j=\bar{1,n}$, $\sum_{i=1}^{m}\sum_{j=1}^{n}h_{ij}\leq A\geq \left\lceil W/\Delta \right\rceil$. The step Δ is chosen to ensure sufficient variability of the resulting integer values of the stochastic elements of the matrix $H=\left(h_{ij}\right)$, $i=\bar{1,m}$, $j=\bar{1,n}$.

Let us formalize the process of forming the values of the elements of the matrix *H*. Let us have *A* atomic units of the resource, which are distributed among *m* × *n* elements of the matrix *H*, and the probability of a resource unit falling into the element $h_{ij}$ is characterized by the probability $p_{ij}$, $i=\bar{1,m}$, $j=\bar{1,n}$. The probability distribution of such a process is defined as
(5)$$P\left(H\right)=A!\prod_{i=1}^{m}\prod_{j=1}^{n}\frac{p_{ij}^{h_{ij}}}{h_{ij}!}.$$

If the Moivre–Stirling approximation of factorials of large numbers is applied to the logarithmic representation of expression (5), we obtain an expression that characterizes the process described above based on the relative entropy:(6)$$E\left(H\left|P\right.\right)=-\sum_{i=1}^{m}\sum_{j=1}^{n}h_{ij}\ln\frac{h_{ij}}{p_{ij}},$$
where $P=\left(p_{ij}\right)$, $i=\bar{1,m}$, $j=\bar{1,n}$.

Taking into account the proposed physical interpretation of the process of the matrix *H* values formation, it is appropriate to introduce such a characteristic parameter as the resource units a priori distribution, i.e., $V=\left(v_{ij}=p_{ij}A\right)$, $i=\bar{1,m}$, $j=\bar{1,n}$. Taking this parameter into account, expression (6) can be redefined as
(7)$$E\left(H\left|V\right.\right)≜-\sum_{i=1}^{m}\sum_{j=1}^{n}h_{ij}\ln\frac{h_{ij}}{v_{ij}}.$$

Equality (7) is defined with accuracy up to the constant *A*ln*A*. The essential connection between the sources of origin of the elements of the matrices *X* and *H* allows us to define the cross-entropy function as
(8)$$E\left(X\left|V\right.\right)≜-\sum_{i=1}^{m}\sum_{j=1}^{n}x_{ij}\ln\frac{x_{ij}}{v_{ij}}.$$

Based on expression (8), we write:(9)$$E\left(G\left|P\right.\right)=-W\ln\frac{W}{A}-\sum_{i=1}^{m}\sum_{j=1}^{n}g_{ij}\frac{g_{ij}}{p_{ij}},$$
where $g_{ij}=x_{ij}/W\in \left[0,1\right]$, $i=\bar{1,m}$, and $j=\bar{1,n}$, and the second term is the relative uncertainty characteristic of the stochastic matrix *X*.

Function (8) is concave for the entire range of values of the argument *X* and reaches a single extremum at the point $x_{ij}^{*}=v_{ij}/e$, e=2,718, $i=\bar{1,m}$, $j=\bar{1,n}$. The extreme value of function (8) is equal to
(10)$$E_{\max}\left(x^{*}\left|V\right.\right)=\frac{1}{e}\sum_{i=1}^{m}\sum_{j=1}^{n}v_{ij}.$$

The value (10) characterizes the maximum uncertainty of the matrix *X* for a defined matrix *V*. Let us emphasize other useful properties of function (8).

Let us define a matrix *L* with elements $l_{ij}\left(x_{ij},v_{ij}\right)=\ln\left(x_{ij}/ev_{ij}\right)$, $i=\bar{1,m}$, and $j=\bar{1,n}$. Considering $V=L\left(X,V\right)$, expression (8) can be rewritten as
(11)$$E\left(X\left|V\right.\right)=E\left(X,L\left(X,V\right)\right)=-\sum_{i=1}^{m}\sum_{j=1}^{n}x_{ij}l_{ij}\left(x_{ij},v_{ij}\right)=\mathrm{Sp}\left(XL^{T}\left(X,V\right)\right)=\mathrm{Sp}\left(L\left(X,V\right)X^{T}\right),$$
where the symbols Sp and T represent the operations of trace finding and matrix transposition, respectively.

Based on the definition $l_{ij}\left(x_{ij},v_{ij}\right)$, we obtain the following inequality for the logarithmic function:(12)$$l_{ij}\left(x_{ij},v_{ij}\right)\leq \left(x_{ij}-v_{ij}\right)/v_{\min},\;i=\bar{1,m},\;j=\bar{1,n},$$
where $v_{\min}=\underset{i,j}{\min}v_{ij}$.

Having transformed expression (11) and taking into account inequality (12), we determine the upper limit of cross entropy (8):(13)$$\hat{E}\left(X\left|V\right.\right)=\mathrm{Sp}\left(XX^{T}\right)-\mathrm{Sp}\left(XV^{T}\right).$$

Function (13) is concave and follows all the properties of function (8).

Consider a non-degenerate $\left(\det\left(V_{\left(n\times m\right)}^{T}V_{\left(m\times n\right)}\right)\neq 0\right)$ matrix of empirical data $V_{\left(m,n\right)}$ with positive elements. Let us set the desired dimension of the parametric space: *r*, *r* < *n*, and enter into the matrix $Q=\left(q_{ij}\geq 0\right)$, $i=\bar{1,n}$, and $j=\bar{1,r}$. We obtain a direct projection of the matrix $Q_{\left(n\times r\right)}$ onto the parametric space *R^mr^*: $Y_{\left(m\times r\right)}=V_{\left(m\times n\right)}Q_{\left(n\times r\right)}$. We obtain the inverse projection on the space *R^mn^* using the matrix $S_{\left(r\times n\right)}$, and the values of all elements which are positive: $X_{\left(m\times n\right)}=V_{\left(m\times n\right)}Q_{\left(n\times r\right)}S_{\left(r\times n\right)}$. The dimensionality of both the obtained matrix *X* and the original matrix *V* is the same: (*m* × *n*).

Let us express the cross-entropy functional $E\left(X\left|V\right.\right)=E\left(X_{\left(m\times n\right)}\left|V_{\left(m\times n\right)}\right.\right)$, taking into account the existence of the matrices $Q_{\left(n\times r\right)}$ and $S_{\left(r\times n\right)}$:(14)$$E\left(X\left|V\right.\right)=E\left(Q,S\left|V\right.\right)=E\left(Q_{\left(n\times r\right)},S_{\left(r\times n\right)}\left|V_{\left(m\times n\right)}\right.\right)=-\sum_{i=1}^{m}\sum_{j=1}^{n}e_{ij}\left(Q_{\left(n\times r\right)},S_{\left(r\times n\right)}\left|V_{\left(m\times n\right)}\right.\right),$$
where
$$e_{ij}\left(Q_{\left(n\times r\right)},S_{\left(r\times n\right)}\left|V_{\left(m\times n\right)}\right.\right)=x_{ij}\left(Q_{\left(n\times r\right)},S_{\left(r\times n\right)}\left|V_{\left(m\times n\right)}\right.\right)\ln\left(x_{ij}\left(Q_{\left(n\times r\right)},S_{\left(r\times n\right)}\left|V_{\left(m\times n\right)}\right.\right)/v_{ij}\right),$$
$$x_{ij}\left(Q_{\left(n\times r\right)},S_{\left(r\times n\right)}\left|V_{\left(m\times n\right)}\right.\right)=\sum_{k=1}^{r}\sum_{l=1}^{n}s_{kj}q_{lk}v_{il},i=\bar{1,m},j=\bar{1,n}.$$

The optimal configuration of the values of the positive matrices *Q* and *S* in the entropy basis is described by the expression
(15)$$\left(Q^{*},S^{*}\right)=\arg\underset{\left(Q,S\right)\geq 0}{\max}E\left(Q,S\left|V\right.\right).$$

We will search for the extremum of the objective function (15) by the iterative gradient projection method [36,37], taking into account the need to cut off elements with negative values (observing condition $\left(Q,S\right)\geq 0$).

Let us analytically express the partial derivatives of the function $E\left(Q,S\left|V\right.\right)$ in terms of the arguments, i.e., the elements of matrices *Q* and *S*:(16)$$\frac{\partial E\left(Q,S\left|V\right.\right)}{\partial q_{kl}}=-\sum_{i=1}^{m}\sum_{j=1}^{n}\frac{\partial e_{ij}\left(Q,S\left|V\right.\right)}{\partial x_{ij}}\frac{\partial x_{ij}\left(Q,S\left|V\right.\right)}{\partial q_{kl}}$$
(17)$$\frac{\partial E\left(Q,S\left|V\right.\right)}{\partial s_{lh}}=-\sum_{i=1}^{n}\sum_{j=1}^{m}\frac{\partial e_{ij}\left(Q,S\left|V\right.\right)}{\partial x_{ij}}\frac{\partial x_{ij}\left(Q,S\left|V\right.\right)}{\partial s_{lh}}$$
where $\partial e_{ij}\left(Q,S\left|X\right.\right)/\partial x_{ij}=\ln\left(x_{ij}/v_{ij}\right)+1$, $\partial x_{ij}\left(Q,S\left|V\right.\right)/\partial q_{kl}=s_{lj}v_{ik}$, $\partial x_{ij}\left(Q,S\left|V\right.\right)/\partial s_{lh}=\sum_{k=1}^{n}q_{kl}v_{ih}$, $i=\bar{1,m}$, $j=\bar{1,n}$, $k=\bar{1,n}$, $l=\bar{1,r}$, and $h=\bar{1,n}$.

Let us derive vectors $\vec{q}$ and $\vec{s}$ as a result vectorization of matrices *Q* and *S*, respectively. We identify the gradient vector of the relative entropy functional (14) with components (16) $\nabla_{Q}\left(\vec{q},\vec{s}\right)$. We identify the gradient vector $\nabla_{S}\left(\vec{q},\vec{s}\right)$ of the relative entropy functional (14) with components (17). We initialize the iterative procedure for finding the extremum of the objective function (15) based on the gradient projection method and in terms of the introduced entities.

For the 0th iteration, we take $X^{\left(0\right)}$, $V^{\left(0\right)}$, ${\vec{q}}^{\left(0\right)}>0$, ${\vec{s}}^{\left(0\right)}>0$.

For the *n*th iteration, we write:(18)$$\begin{array}{c} {\vec{q}}^{\left(n+1\right)}=\left\{\begin{array}{l} {\vec{q}}^{\left(n\right)}+\gamma_{\vec{q}}\nabla_{Q}\left({\vec{q}}^{\left(n\right)},{\vec{s}}^{\left(n\right)}\right)\forall {\vec{q}}^{\left(n+1\right)}\geq 0,\\ {\vec{q}}^{\left(n\right)}\forall {\vec{q}}^{\left(n+1\right)}<0, \end{array}\right.\\ {\vec{s}}^{\left(n+1\right)}=\left\{\begin{array}{l} {\vec{s}}^{\left(n\right)}+\gamma_{\vec{s}}\nabla_{S}\left({\vec{q}}^{\left(n\right)},{\vec{s}}^{\left(n\right)}\right)\forall {\vec{s}}^{\left(n+1\right)}\geq 0,\\ {\vec{s}}^{\left(n\right)}\forall {\vec{s}}^{\left(n+1\right)}<0, \end{array}\right.\\ {\vec{q}}^{\left(n+1\right)}\Rightarrow Q^{\left(n+1\right)},{\vec{s}}^{\left(n+1\right)}\Rightarrow S^{\left(n+1\right)},X^{\left(n+1\right)}=Q^{\left(n+1\right)}S^{\left(n+1\right)}V,\\ E^{\left(n+1\right)}=E\left(Q^{\left(n+1\right)},S^{\left(n+1\right)}\left|V\right.\right)=\sum_{i=1}^{m}\sum_{j=1}^{r}x_{ij}^{\left(n+1\right)}\ln\frac{x_{ij}^{\left(n+1\right)}}{v_{ij}}, \end{array}$$
where parameters $\gamma_{\vec{q}}$, $\gamma_{\vec{s}}$ regulate increments in the corresponding dimension.

Iterative process (18) ends when the dynamics of the change in the value of the relative entropy functional becomes less than the threshold *δ*:(19)$$\delta_{E}=E^{\left(n+1\right)}-E^{\left(n\right)}=\frac{I\left(V\right)-I\left(Y\left(Q\left|V\right.\right)\right)}{I\left(V\right)}\leq \delta ,$$
where $I\left(V\right)=\sum_{i=1}^{m}\sum_{j=1}^{n}v_{ij}\ln v_{ij}$ is the information capacity of the positive matrix $V_{\left(m\times n\right)}$. By analogy, we write: $I\left(Y\left(Q\left|V\right.\right)\right)=\sum_{i=1}^{m}\sum_{j=1}^{r}y_{ij}\left(Q\left|V\right.\right)\ln y_{ij}\left(Q\left|V\right.\right)$, where $y_{ij}=\sum_{l=1}^{n}v_{il}q_{lj}$.

The computational complexity of the implementation of the iterative procedure just described increases nonlinearly with the increase in the dimension of the analyzed empirical matrices. Considering this circumstance, it is acceptable to define the elements of the matrix of the reduced dimension *Q* based on the approximately defined relative entropy functional $\tilde{E}$. For example, let us use the approximation of the logarithmic function at the point $x_{0}=w$: $\ln x<\ln w+\left(x-w\right)/w_{\min}$. For points $w=x_{ij}$ we find:(20)$$E\left(Q,S\left|V\right.\right)\approx \tilde{E}\left(Q,S\left|V\right.\right)=\sum_{i=1}^{n}\sum_{j=1}^{m}\left(x_{ij}^{2}\left(Q,S\left|V\right.\right)-x_{ij}\left(Q,S\left|V\right.\right)v_{ij}\right).$$

Let us present the expression (20) in the matrix form:(21)$$\tilde{E}\left(Q,S\left|V\right.\right)=\mathrm{Sp}\left(XX^{T}\right)-\mathrm{Sp}\left(XV^{T}\right),=†(X(Q,S),X(Q,S))-†(X(Q,S),V)$$
where the symbol † represents the Frobenius scalar product: $\mathrm{Sp}\left(AB^{T}\right)=\mathrm{Sp}\left(BA^{T}\right)=†\left(A,B\right)=†\left(B,A\right)$.

With a fixed matrix of empirical data *V*, we will minimize the functional $\tilde{E}\left(Q,S\left|V\right.\right)$ on the set of positive matrices *Q* and *S*:(22)$$\left({\tilde{Q}}^{*},{\tilde{S}}^{*}\right)=\arg\underset{\left(Q,S\right)\geq 0}{\min}\tilde{E}\left(Q,S\left|V\right.\right).$$

The procedure for finding $\left({\tilde{Q}}^{*},{\tilde{S}}^{*}\right)$ also uses components (16), and (17), which should be adapted to the scalar form of representation of the entities involved. Applying the rules of matrix differentiation to the functional (21), we obtain the following scalar interpretations of components (16), and (17):(23)$$\Delta_{Q}\left(Q,S\right)=\frac{\partial \tilde{E}\left(Q,S\left|V\right.\right)}{\partial X}\frac{\partial X}{\partial Q}=2SXQX-SX,$$
(24)$$\Delta_{S}\left(Q,S\right)=\frac{\partial \tilde{E}\left(Q,S\left|V\right.\right)}{\partial X}\frac{\partial X}{\partial S}=2Q^{T}XQX-Q^{T}X,$$
where $X=XX^{T}$; $\Delta_{Q}\left(Q,S\right)$ and $\Delta_{S}\left(Q,S\right)$ are the gradients of matrices *Q* and *S*, respectively. The results of expressions (23), and (24) will be matrices of dimension (*n* × *r*).

We initialize the iterative procedure for finding the extremum of objective function (22) based on the gradient descent method and in terms of entities (23), and (24).

For the 0th iteration: we take *X*^(0)^, *V*^(0)^.

For the *n*th iteration, we write:(25)$$\begin{array}{c} Q^{\left(n+1\right)}=\left\{\begin{array}{l} Q^{\left(n\right)}+\gamma_{Q}\Delta_{Q}\tilde{E}\left(Q^{\left(n\right)},S^{\left(n\right)}\left|V\right.\right)\geq 0\forall Q^{\left(n+1\right)}\geq 0,\\ Q^{\left(n\right)}\forall Q^{\left(n+1\right)}<0, \end{array}\right.\\ S^{\left(n+1\right)}=\left\{\begin{array}{l} S^{\left(n\right)}+\gamma_{S}\Delta_{S}\tilde{E}\left(Q^{\left(n\right)},S^{\left(n\right)}\left|V\right.\right)\geq 0\forall S^{\left(n+1\right)}\geq 0,\\ S^{\left(n\right)}\forall S^{\left(n+1\right)}<0, \end{array}\right.\\ X^{\left(n+1\right)}=VQ^{\left(n+1\right)}S^{\left(n+1\right)},E^{\left(n+1\right)}=\sum_{i=1}^{m}\sum_{j=1}^{r}x_{ij}^{\left(n+1\right)}\ln\frac{x_{ij}^{\left(n+1\right)}}{v_{ij}}. \end{array}$$

The iterative process (25) ends when the dynamics of the change in the value of the functional $\tilde{E}\left(Q,S\left|V\right.\right)$ becomes less than the set threshold *δ*: $E^{\left(n+1\right)}-E^{\left(n\right)}\leq \delta$.

In [35], the authors described the basic concept of solving the *d*- and *c*-problems mentioned in Section 2.1 for empirical data of the type *V* and *Y*, respectively. The result is the optimal probability distribution densities of characteristic parameters and interference (for the *d*-problem: $P^{*}\left(w\right)$, $L^{*}\left(\varepsilon \right)$, and for the *c*-problem: $A^{*}\left(a\right)$, $Z^{*}\left(\xi \right)$, respectively). The mathematical apparatus presented in Section 2.2 allows, based on linear models (2), and (4), to calculate normalized U∩S probability distributions $F_{d}\left(\vec{u}\right)$ and $F_{c}\left(\vec{s}\right)$ to determine the absolute difference between these functions in terms of relative entropy [38,39,40,41].

To preserve the integrity of the presentation of the material, we will demonstrate how the basic concept of solving the *d*-problem is implemented in the context of model (2). Let’s define the functional $E\left(P\left(w\right),L\left(\varepsilon \right)\right)$ on the probability distribution densities $P^{*}\left(w\right)$ and $L^{*}\left(\varepsilon \right)$. We need to solve the optimization problem with the following objective function and constraints:(26)$$\begin{array}{l} E\left(P\left(w\right),L\left(\varepsilon \right)\right)=-\int_{W}P\left(w\right)\ln P\left(w\right)dw-\int_{E}L\left(\varepsilon \right)\ln L\left(\varepsilon \right)d\varepsilon \to \max\\ \;\int_{W}P\left(w\right)dw=1,\int_{E}L\left(\varepsilon \right)d\varepsilon =1,M\left\{z\right\}=\int_{W}VwP\left(w\right)dw+\int_{E}\varepsilon L\left(\varepsilon \right)d\varepsilon =y. \end{array}$$

The solution to the optimization problem (26) in analytical form looks like
$$P^{*}\left(w\right)=\exp\left(-\theta ,Vw\right)/\int_{W}\exp\left(-\theta ,Vw\right)dw,L^{*}\left(\varepsilon \right)=\exp\left(-\theta ,\varepsilon \right)/\int_{Β}\exp\left(-\theta ,\varepsilon \right)d\varepsilon ,$$
where the Lagrange multipliers *θ* are determined as a result of solving the system of balance equations $M\left\{z\right\}$ in the interpretation $\int_{W}VwP^{*}\left(w\right)dw+\int_{E}\varepsilon L^{*}\left(\varepsilon \right)d\varepsilon =y$.

In the context of the model (2), the probability distribution density *F*(*u*) of the observation vector *u* is defined as
(27)$$F\left(u\right)=\int_{E}\Pi \left(u-\varepsilon \right)L^{*}\left(\varepsilon \right)d\varepsilon =F_{d}\left(u\right),$$
where *F_d_*(*u*) is the desired probability distribution density of the *d*-problem model, and $\Pi \left(u-\varepsilon \right)$ is the density of the stochastic vector u−ε. From expression (27) we find $w=V^{T}z/V^{T}V$.

Considering the interval nature of the vector *z*: $z\in Z=\left[z^{-}=Vw^{-},z^{+}=Vw^{+}\right]$, we write $\eta \left(z\right)=P^{*}\left(V^{T}z/V^{T}V\right)$. Having normalized the function $\eta \left(z\right)$, we express the probability distribution density of the vector *z* as $\Pi \left(z\right)=\eta \left(z\right)/\int_{Z}\eta \left(z\right)dz$.

To determine the probability distribution density $F_{c}\left(s\right)$ in the context of the model (4) (*c*-problem), it is necessary to repeat the sequence of actions embodied in expression (27) based on the empirical data matrix *Y*.

To compare the functions $F_{d}\left(u\right)$ and $F_{c}\left(s\right)$, it is necessary to normalize them on the common carrier Λ=U∩S:(28)$${\tilde{F}}_{d}\left(\lambda \right)=F_{d}\left(\lambda \right)/\int_{\Lambda }F_{d}\left(\lambda \right)d\lambda ,\;{\tilde{F}}_{c}\left(\lambda \right)=F_{c}\left(\lambda \right)/\int_{\Lambda }F_{c}\left(\lambda \right)d\lambda .$$

To find the absolute share of information losses between functions ${\tilde{F}}_{d}\left(\lambda \right)$ and ${\tilde{F}}_{c}\left(\lambda \right)$ due to compaction Δ_*E*_ we define in terms of the relative entropy of *RE* as
$$RE\left({\tilde{F}}_{d},{\tilde{F}}_{c}\right)=\int_{\Lambda }{\tilde{F}}_{c}\left(\lambda \right)\ln\left({\tilde{F}}_{c}\left(\lambda \right)/{\tilde{F}}_{d}\left(\lambda \right)\right),$$
(29)$$\Delta_{E}=\frac{1}{2}RE\left({\tilde{F}}_{d},{\tilde{F}}_{c}\right)+RE\left({\tilde{F}}_{d},{\tilde{F}}_{c}\right).$$

Note that the minimum Δ_*E*_ = 0 is reached at ${\tilde{F}}_{d}\left(\lambda \right)={\tilde{F}}_{c}\left(\lambda \right)$.

## 3. Results

Let us begin the experimental Section by demonstrating the functionality of the mathematical apparatus proposed in Section 2.2 on a simple abstract example.

Suppose we have initial empirical data of the form $V_{\left(m=2\times n=2\right)}=\left(\begin{array}{cc} 0,100 & 0,800\\ 0,800 & 1,000 \end{array}\right)$. In the context of model (2), we write u=Vw+ε. Suppose that $w\in W=\left[0,000;5,000\right]$, $\varepsilon \in E=\left[-0,500;0,500\right]$. The output component is defined by the vector $y=\left(0,600;1,400\right)$.

Let r=1, then $Y_{\left(2\times 1\right)}=V_{\left(2\times 2\right)}Q_{\left(2\times 1\right)}$, where $Q_{\left(2\times 1\right)}=\left(\begin{array}{c} q_{11}\\ q_{21} \end{array}\right)$ is the matrix for the direct projection. The compactification model (4) for the above values and conditions looks like this s=Ya+ξ, where $a\in a={\left[0,000;5,000\right]}_{}^{}$, $\xi \in \Xi =\left[-0,500;0,500\right]$. The inverse projection operation is analytically characterized as $X_{\left(2\times 2\right)}=V_{\left(2\times 2\right)}Q_{\left(2\times 1\right)}S_{\left(1\times 2\right)}$, where $S_{\left(1\times 2\right)}=\left(\begin{array}{cc} s_{11} & s_{12} \end{array}\right)$ is the matrix for the inverse projection.

Our example is characterized by a small dimension, so we will use procedure (18) to determine the cross entropy. In this context, the cross entropy *E* between the original empirical matrix $V_{\left(2\times 2\right)}$ and the matrix $X_{\left(2\times 2\right)}$ obtained as a result of direct-inverse projection will be analytically determined by the expression $E=-\sum_{i=1}^{2}\sum_{j=1}^{2}x_{ij}\ln\frac{x_{ij}}{v_{ij}}$. The function $E\left(Q,S\right)$ reaches an extremum at $Q_{\max}^{*}=\left(\begin{array}{c} 0,356\\ 0,768 \end{array}\right)$, $S_{\max}^{*}=\left(\begin{array}{cc} 0,257 & 0,559 \end{array}\right)$. Accordingly, the optimal compactified matrix Y has the form $Y^{*}=\left(\begin{array}{c} 0,356\\ 0,768 \end{array}\right)$.

The optimal probability distribution densities of the characteristic parameters w and interference ε for the matrix *V* defined at the beginning of the Section are characterized by the functions $P^{*}\left(w\right)=1,221\exp\left(-0.888w_{1}-1,419w_{2}\right)$ and $L^{*}\left(\varepsilon \right)=0,982\exp\left(-0,642\varepsilon_{1}-0,136\varepsilon_{2}\right)$. To compare the functions $F_{d}\left(u\right)$ and $F_{c}\left(s\right)$, it is necessary to normalize them on the common carrier, so, using (28), we find $0\leq \lambda_{1}\leq 1,778$, $0\leq \lambda_{2}\leq 3,842$. Then, with the defined functions (2), (4), and $P^{*}\left(w\right)$, the absolute share of information losses (29) of reducing the dimensionality of the space of characteristic parameters from *n* = 2 to *r* = 1 $\left(V_{\left(2\times 2\right)}\to Y_{\left(2\times 1\right)}^{*}\right)$ is equal to $\Delta_{E}=0,245$, which allows us to consider the result of the proposed compactification procedure of the original empirical matrix *V* as adequate.

To prove the effectiveness of the proposed compactification method (18) (*Met*3), the method should be compared with popular analogues, namely, with the principal component analysis method (*Met*1) and the random projection method (*Met*2). Considering the linear nature of functions (2) and (4), we will experiment in the context of solving the verification problem (dichotomous classification) with a linear classifier. Let’s formulate such a problem based on the terminology used.

We define the linear classifier model as
(30)$$z\left(s_{k}\right)=\mathrm{sign}\left(\sum_{i=1}^{n}w_{i}v_{i}\left(s_{k}\right)\right)=\left\{\begin{array}{l} +1\forall \sum_{i=1}^{n}w_{i}v_{i}\left(s_{k}\right)\geq 0,\\ -1\forall \sum_{i=1}^{n}w_{i}v_{i}\left(s_{k}\right)<0, \end{array}\right.$$
where $k\in \left\{1,m\right\}$ and the values of the weights $w\in R^{n}$ are unknown a priori.

Empirical data with the structure $\langle V_{\left(m\times n\right)},y_{\left(m\times 1\right)}\rangle$ are available, and $y_{k}=\left\{\begin{array}{l} +1\forall z\left(t_{k}\right)=+1,\\ -1\forall z\left(t_{k}\right)=-1, \end{array}\right.$ where $t_{k}$ is an instance of a class $\langle V,y\rangle$ with a number $k\in \left\{1,m\right\}$. The training of the classifier (30) is reduced to the minimization of the empirical risk function of the form $R\left(w\right)=\sum_{i=1}^{m}{\left\|y-z\left(w\left|V\right.\right)\right\|}^{2}$. To test the trained classifier (30), test empirical data with the structure $\langle U_{\left(l\times n\right)},x_{\left(l\times 1\right)}\rangle$ were used.

The results of the classification $b\left(t_{k}\right)=sign\left(\sum_{i=1}^{n}{\hat{w}}_{i}u_{i}\left(t_{k}\right)\right)=\left\{-1,1\right\}\forall k=\bar{1,l}$ are synchronously compared with the corresponding elements of the vector x and taken into account in the form of the value of the function $I=\sum_{k-1}^{l}\Delta \left(t_{k}\right)$, where $\Delta \left(t_{k}\right)=\left\{\begin{array}{l} 1\forall b\left(t_{k}\right)=x\left(t_{k}\right),\\ 0\forall b\left(t_{k}\right)\neq x\left(t_{k}\right). \end{array}\right.$. Accordingly, classification accuracy is defined as α=I/l.

The conducted experiment consisted of solving the verification problem using classifier (30) for:

*e*0—basic empirical dataset $\langle V_{\left(m\times n\right)},y_{\left(m\times 1\right)}\rangle +\langle U_{\left(l\times n\right)},x_{\left(l\times 1\right)}\rangle$;

{*e*1, *e*2, *e*3}—the dataset $\langle V,y\rangle +\langle U,x\rangle$, the dimension of the attributes of the matrices *V* and *U* which underwent compactification from the initial *n* to the specified *r* elements by the method {*Met*1, *Met*2, *Met*3}.

The value r was iteratively reduced: r=n−1,n−2,…,1, forming a set of datasets at each of the stages {*e*1, *e*2} with the corresponding compactification degree. The number of compactification procedures *e*3 was determined by the set of threshold values (19).

For experiments, as necessary, tables of synthetic data of the required size were generated. For this, the *sklearn.datasets.make_classification*(*n_class* = 2, *n_clusters* = 2, *n_redundant* = 0, *class_sep* = 1.0, *n_informative* = {10, 15}) function of the *Python* programming language was used. Before use, all generated data were normalized to fall within the unit interval [0, 1]. The experiments were carried out using *scipy.stats.bootstrap* cross-validation.

The algorithmic designs of the {*Met*1, *Met*2, *Met*3} methods were implemented by the functions of the *scipy* and *sklearn* libraries. The classifier (30) was implemented as a support vector machine with a linear kernel using the function *sklearn.svm.SVC*. The *Met*1 *Met*2 methods were implemented using the *sklearn.decomposition.PCA* and *sklearn.random_projection.GaussianRandomProjection* functions, respectively. The basis for the implementation of the author’s method (18) was the *scipy.optimize.minimize* function (after inverting the objective function (15)). At the same time, the attribute *ftol* was considered to be related to the threshold (19).

As already mentioned in Section 2.2, the author’s empirical data compactification method proposed in the form of procedure (18) is comparatively computationally complex (this is what prompted the authors to formalize the “simplified” iterative procedure (25)). However, *Met*1, *Met*2 analogues have their disadvantages, which appear when compacting large-dimensional data. For example, with a sufficiently large number of instances of data *m* and their heterogeneity, *Met*1 becomes unstable. We will conduct the first experiment of the form *α* = *f*(*m*, *Met*, *r*) for $m=\bar{5,10}$, n=10, $r=\bar{10,5}$, *Met* = {*Met*1, *Met*2, *Met*3}. The obtained results are visualized in Figure 1.

The previous experiment characterized the ultra-compact empirical data compactification procedure: m≈n, n/2≤r≤n. Now, let us investigate how the verification accuracy *α* depends on the compactification of the initial data, for which m>>r, m>n. The experiments were carried out for two generated datasets *DS*1 and *DS*2. The first was characterized by dimension (*m* = 100, *n* = 10) and the second by dimension (10^4^, 10^2^). When processing the first dataset, we set *r* = {100, 90,…, 50} When working with the second dataset, we set *r* = {100, 90,…, 50} The obtained results are presented in Figure 2.

The following experiment is specific to *Met*3 because it concerns the detection of the dependence between the verification accuracy *α* and the dynamics of such parameters as the compactification degree *r* and the value of the threshold $\delta =\left\{0,5;0,4;\ldots ;0,1\right\}$ (see expression (19)). To preserve the common information background, the remaining parameters were borrowed from the previous experiment without changes, namely: $DS=\left\{DS1,DS2\right\}$, *r*(*DS*1) = {10, 9,…, 5}, and *r*(*DS*2) = {100, 90,…, 50}. The resulting dependencies are visualized in Figure 3.

The empirical data compactification process is accompanied by an information loss. The absolute error as an indicator of information loss during compactification can be calculated by expression (29). The relative share of information losses during compactification can be calculated directly by expression (19) when implementing the compactification procedure (18). Figure 4 presents the calculated dependences of the relative share of information loss $\delta_{E}$ on the compactification method $Met=\left\{Met1,Met2,Met3\right\}$ for datasets $\left\{DS1,DS2\right\}$ with the corresponding ranges of changes in the compaction degree r.

Finally, we will conclude the Experimental Section with a study of *Met*3, the detection of the dependence between the relative share of information loss $\delta_{E}$, and the dynamics of such parameters as the compactification degree r and the threshold value $\delta =\left\{0,5;0,4;\ldots ;0,1\right\}$ (see expression (19)). To ensure a holistic perception of the material of the Section, the remaining parameters were borrowed from the previous experiment. The resulting dependencies are shown in Figure 5.

## 4. Discussion

The research subject was chosen to reveal the characteristic features of the research object. This axiom works in all areas of science. Data analysis is no exception. There can be a huge, large, or small amount of data. The case with a small amount of data may be complicated by the fact that the source of the data, the process of its collection, or both, may not be under the researchers’ control. In this case, data scientists will have to work with small stochastic data. The mathematical apparatus presented in Section 2.2 is focused on the problem of analyzing such data. Objective functions (15), and (22) implement the principle of maximum entropy formulated by Willard Gibbs in the context of compactification of (small) stochastic empirical data. Gibbs’ work says that the most characteristic probability distributions of the states of an uncertain object are distributions that maximize the chosen measure of uncertainty, taking into account the available reliable information about the investigated object. The effectiveness of this approach is demonstrated by the results presented in Figure 1. Recall that, in this experiment, the compactification of extremely small data was carried out (the number of instances m in the data collection approached the number of attributes *n*). From Figure 1a,b, it can be seen that both the principal component analysis method (*Met*1) and the random projection method (*Met*2) demonstrated cases of non-functionality in situations when *m* < *r*, where *r* was the desired number of attributes in the compactified collection. The author’s method (*Met*3) remained functional under any requirements determined by the experiment.

As shown in Figure 1, the results characterized the small empirical data compactification process: *m* ≈ *n*, n/2≤r≤n, then the results presented in Figure 2 show how the verification accuracy *α* depends on the compactification of the initial data, for which m>n (a sufficient amount of empirical data, Figure 2a) or m>>r (“big” empirical data, Figure 2b). From Figure 2a, it can be seen that r≤7 of function $\alpha \left(Met3\right)$ shows a monotonic linear character, in contrast to functions $\alpha \left(Met1\right)$ and $\alpha \left(Met2\right)$. This circumstance indicates that it was the author’s method that made it possible to find the optimal configuration of the characteristic parameters space. Instead, the change of r in all functions $\alpha \left(Met\right)$ from Figure 2b is characterized by a non-linear character. It can also be seen that, with r≤60, it is the author’s compactification method Met3 that generates the least informative parametric space in comparison with analogues. This fact can be explained by the fact that optimization method (18) does not have time to come close to the optimal distribution ensemble for the maximum number of iterations set of the algorithm (attribute maxiter=1000 for the function scipy.optimize.minimize). The way out in such a situation can be the application of the approximate version of algorithm (18), represented by expressions (25).

Figure 3 demonstrates the dependence of the verification accuracy *α* on the dynamics of such parameters as the compactification degree *r* and the threshold value $\delta =\left\{0,5;0,4;\ldots ;0,1\right\}$ (see expression (19)) of the completion of the iterative procedure (18). Let us notice that threshold δ is also a parameter that determines the maximum allowable reduction of the information capacity for the compactification data matrix. The usefulness of parameter δ lies in the fact that, based on its value, we can choose the permissible compaction degree r, not empirically (as, for example, in Met1) but analytically; if, after reducing the dimensionality of the dimension of the characteristic parameters to the value $r^{\left(n\right)}$, the estimate $\delta_{E}$ has decreased too much, then the compactification process should be stopped and the algorithm should be rolled back to the previous value of $r^{\left(n-1\right)}$. This is exactly the behaviour we observe in Figure 3a. Instead, as shown in Figure 3b, the situation is not stable. The probable explanation for this is similar to the one we mentioned regarding Figure 2b.

Figure 4 presents the calculated dependences of the relative share of information loss $\delta_{E}$ on the compactification method $Met=\left\{Met1,Met2,Met3\right\}$ for datasets $\left\{DS1,DS2\right\}$ with the corresponding ranges of changes in the compactification degree *r*. It can be seen that it is the function $\delta_{E}=f\left(r,Met3\right)$ with the growth *r* that grows significantly more slowly, surpassing competitors by almost two times. Note that this advantage was observed both for the “large” dataset *DS*1 and for the “Big” dataset *DS*2.

Figure 5 shows the relationship formalized by expression (19) between the relative share of information loss $\delta_{E}$ and the dynamics of such parameters as the compactification degree *r* and the threshold value δ. It is interesting that, for the dataset *DS*1 (Figure 5a), all values of *r* the condition $\delta_{E}\leq \delta$ are fulfilled, that is, algorithm (18) managed to find optimal distributions without exceeding the set limit on the permissible number of iterations. On the other hand, the circumstances were different for the “Big” dataset *DS*2. This can explain the unstable nature of the values presented in Figure 5b.

In general, the results presented in Section 3 prove both the functionality and the effectiveness of the mathematical apparatus presented in Section 2 in comparison with classical analogues, namely, the principal component analysis method and the random projection method. The obvious advantage of the author’s method is the demonstrated stability of the small stochastic data compactification process and the possibility of analytical control of the loss of information capacity of the compactification data matrix. On the other hand, the disadvantage of the author’s method is the computational complexity, which is especially evident when processing large data matrices. However, to mitigate this limitation, the authors propose an approximating simplified version (25) of the basic compactification procedure (18).

To implement the cross-entropy version of the author’s compactification method, the method of conditional optimization on a non-negative orthant (CONNO) is adapted, and implemented in the *scipy* library. We note that, for some combinations of input data, the basic version of the CONNO method does not find a solution for the given optimization parameters. To test this concept, a series of experiments were adopted. The first series of experiments focused on identifying the dependence of classification accuracy on the number of objects (i.e., sample size). The study of this dependence for three compactification methods (PCA, RP, and author’s) is important to identify areas of their application. It is known that entropy maximization methods and their derivatives, in particular the author’s method, are usually used when the amount of data is limited compared to the dimension of the feature space. With “Big Data,” there are no fundamental restrictions on their use, but computational difficulties increase significantly. The next series of experiments was focused on identifying the dependence of classification accuracy in conditions where the number of measurements significantly exceeds the number of characteristic parameters. The next series of experiments was focused on identifying the dependence of classification accuracy for the author’s method on the acceptable reduction in the information capacity of the dataset. The next series of experiments was focused on assessing information losses from compactification implemented using and for the author’s method. The experiments described above have already been carried out and results that positively characterize the author’s method have been obtained. The problem is that, in its final form, the description, results obtained, and discussion are already more than 10 pages long. Increasing the size of this (already massive) article does not seem practical; therefore, if the mentioned experimental results interest you, dear reader, then I ask you to contact the corresponding author and he will be happy to share with you the results mentioned above.

## 5. Conclusions

Measurement is a typical way of gathering information about the investigated object, generalized by a finite set of characteristic parameters. The result of each iteration of the measurement is an instance of the class of the investigated object in the form of a set of values of characteristic parameters. An ordered set of instances forms a collection whose dimensionality for a real object is a factor that cannot be ignored. Managing the dimensionality of data collection, as well as classification, regression, and clustering, are fundamental problems of machine learning.

Compactification is the approximation of the original data collection by an equivalent collection (with a reduced dimension of characteristic parameters) with the control of accompanying information capacity losses. Related to compactification is the data completeness verifying procedure, which is characteristic of the data reliability assessment. If there are stochastic parameters among the initial data collection characteristic parameters, the compactification procedure becomes more complicated. To take this into account, the research proposes a model of a structured collection of stochastic data defined in terms of relative entropy. The compactification of such a data model is formalized by an iterative procedure aimed at maximizing the relative entropy of sequential implementation of direct and reverse projections of data collections, taking into account the estimates of the probability distribution densities of their attributes. The procedure for approximating the relative entropy function of compactification to reduce the computational complexity of the latter is proposed. For a qualitative assessment of compactification, the metric of such indicators as the data collection information capacity, and the absolute and relative share of information losses due to compaction, are analytically formalized. Taking into account the semantic connection of compactification and completeness, the proposed metric is also relevant for the data reliability assessment task. Testing the proposed compactification procedure proved both its stability and efficiency in comparison with such used analogues as the principal component analysis method and the random projection method.

Further research is planned to attempt to simplify the procedure for finding entropy-optimal matrix projectors while observing the limit on permissible information losses from compactification.

## Figures and Tables

**Figure 1 entropy-25-01567-f001:**
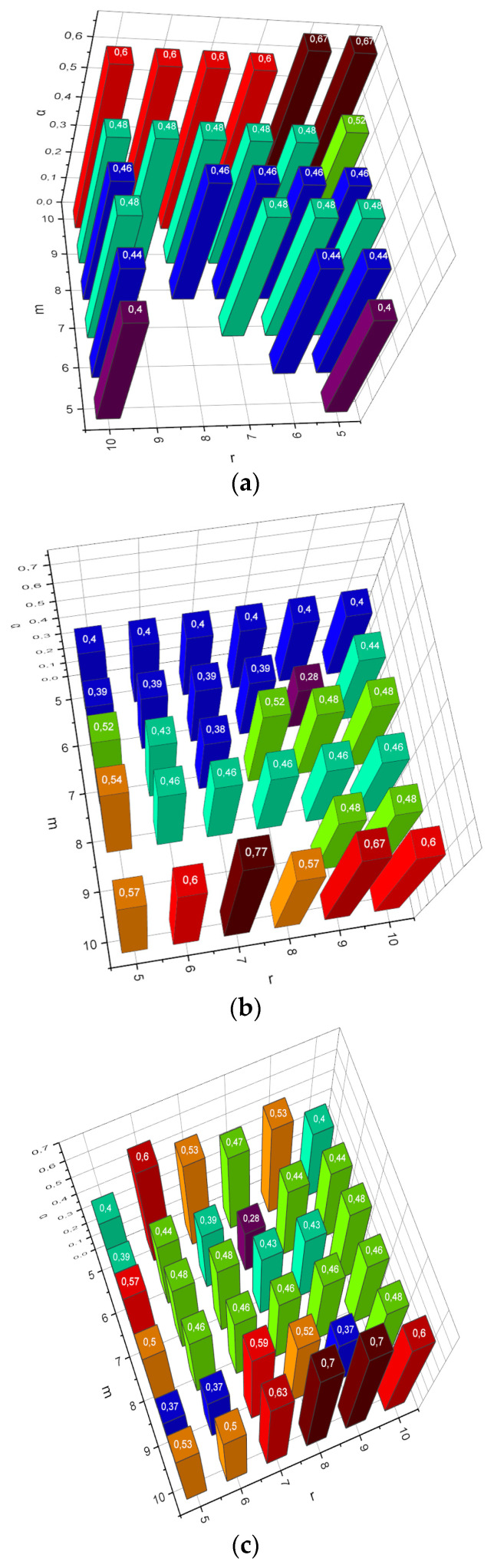
(**a**) Dependence *α* = *f*(*m*, *Met*1, *r*), $m=\bar{5,10}$, $r=\bar{10,5}$. (**b**) Dependence *α* = *f*(*m*, *Met*2, *r*), $m=\bar{5,10}$, $r=\bar{10,5}$. (**c**) Dependence *α* = *f*(*m*, *Met*3, *r*), $m=\bar{5,10}$, $r=\bar{10,5}$.

**Figure 2 entropy-25-01567-f002:**
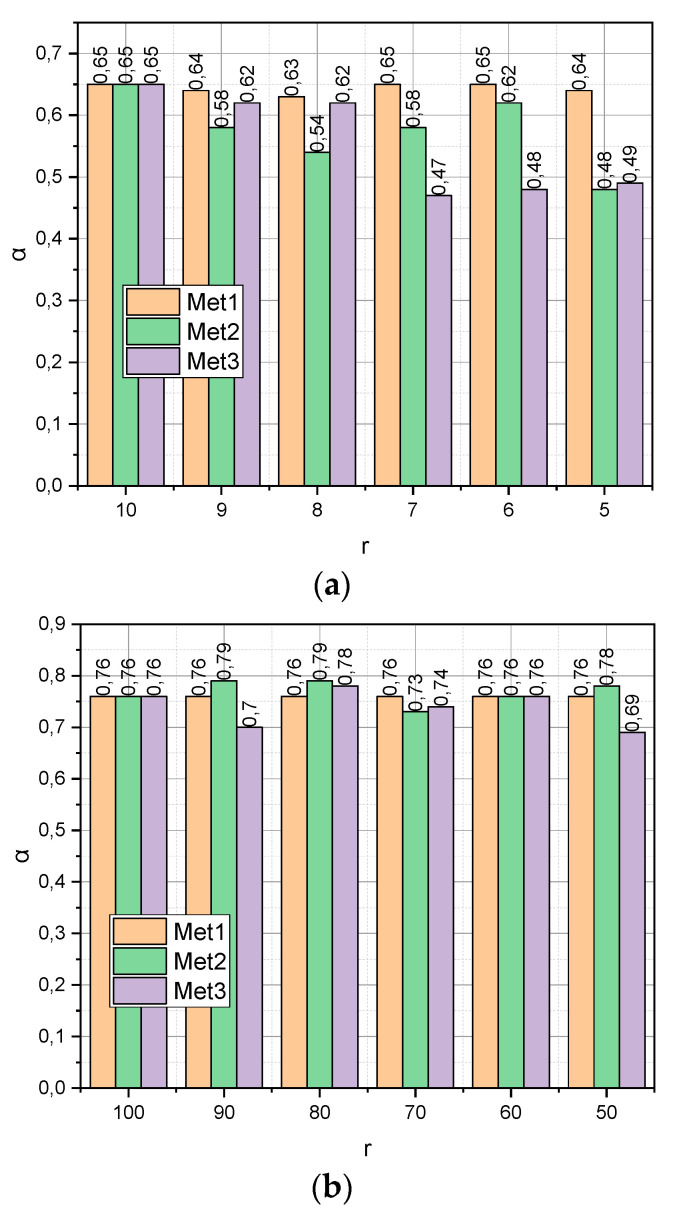
(**a**) Dependence *α* = *f*(*Met*, *r*, *DS*1), $r=\bar{10,5}$, *Met* = {*Met*1, *Met*2, *Met*3}. (**b**) Dependence *α* = *f*(*Met*, *r*, *DS*2), *r* = {100, 90,…, 50}, *Met* = {*Met*1, *Met*2, *Met*3}.

**Figure 3 entropy-25-01567-f003:**
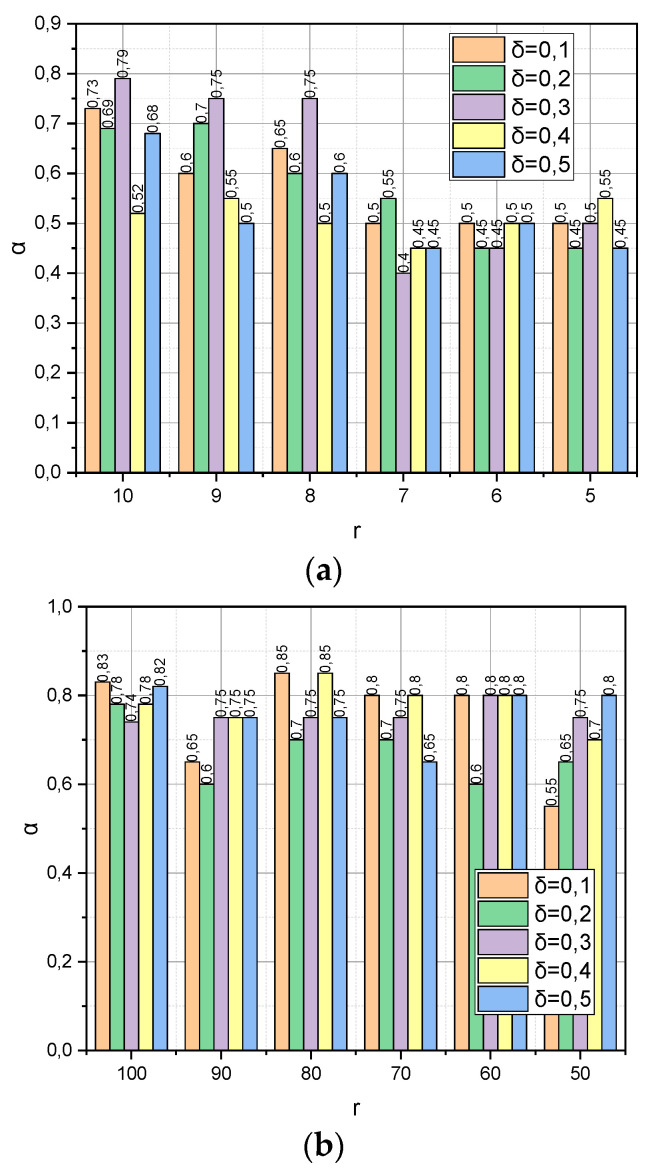
(**a**) Dependence *α* = *f*(*r*, *δ*) for *DS*1 dataset. (**b**) Dependence *α* = *f*(*r*, *δ*) for *DS*2 dataset.

**Figure 4 entropy-25-01567-f004:**
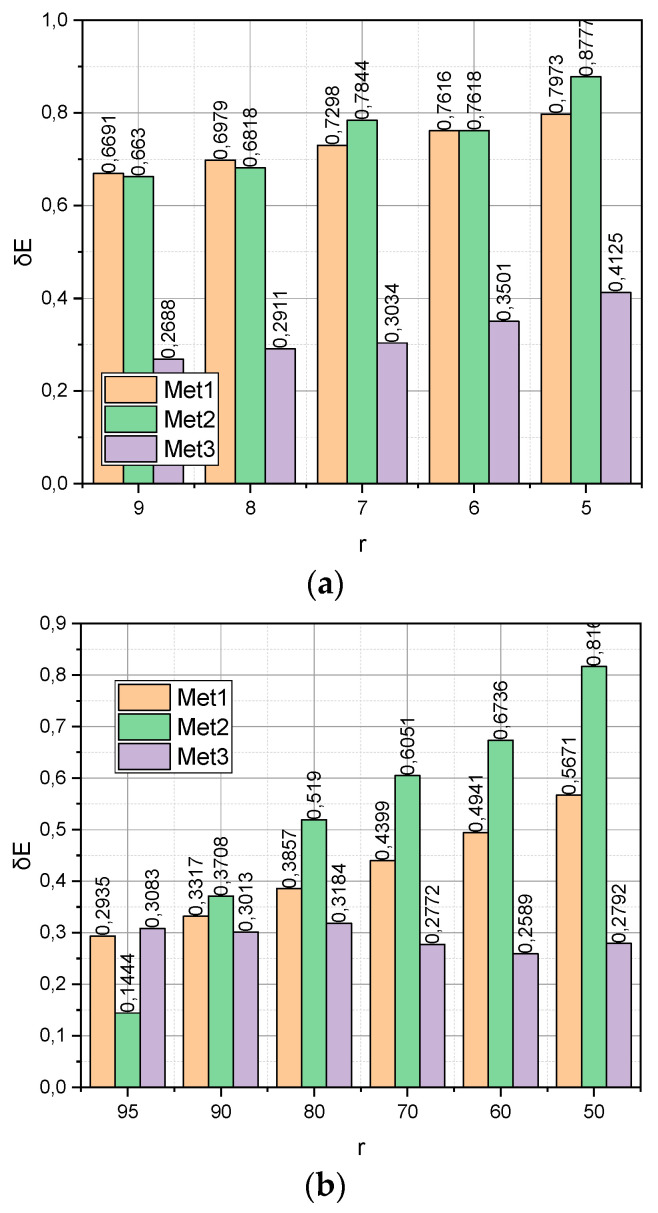
(**a**) Dependence $\delta_{E}=f\left(r,Met\right)$ for DS1 dataset. (**b**) Dependence $\delta_{E}=f\left(r,Met\right)$ on DS2 dataset.

**Figure 5 entropy-25-01567-f005:**
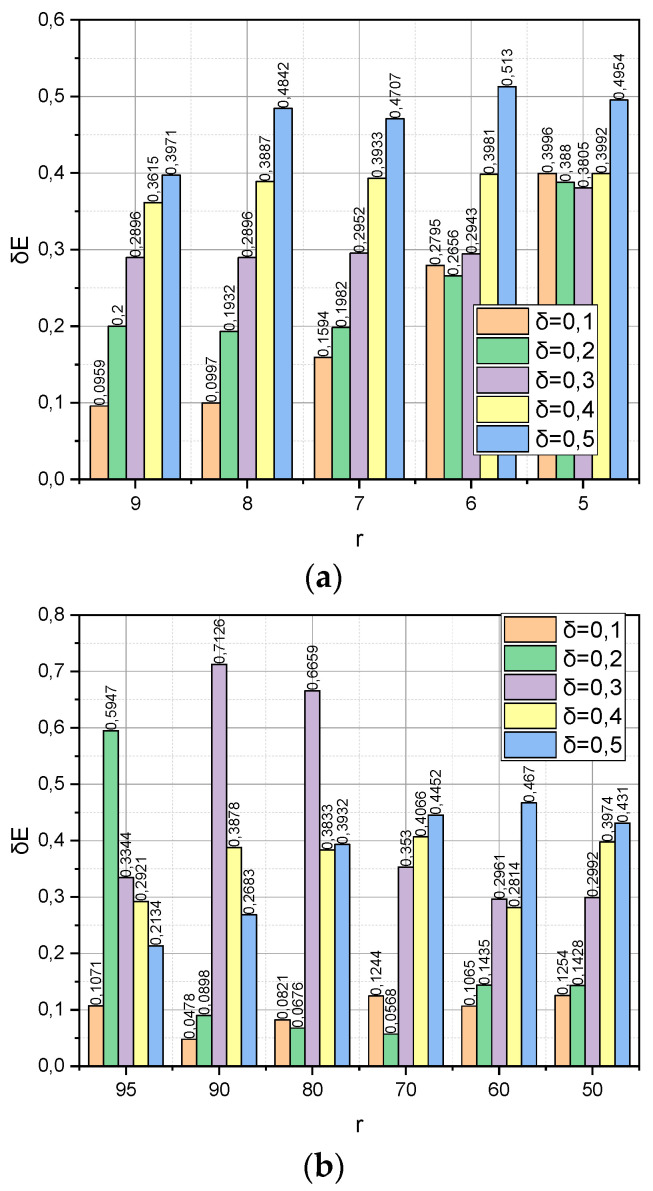
(**a**) Dependence *δ_E_* = *f*(*r*, *δ*) for *Met*3 and *DS*1 dataset. (**b**) Dependence *δ_E_* = *f*(*r*, *δ*) for *Met*3 and *DS*2 dataset.

**Table 1 entropy-25-01567-t001:** General comparison of the concepts of machine and deep learning.

Criterion	Machine Learning	Deep Learning
The number of data points	One can use small amounts of data to create forecasts	It is necessary to use large volumes of training data to create forecasts
Dependence on equipment	It can work on low-power computers. Large computing power is not required	Depends on high-performance computers. At the same time, the computer performs a large number of operations on the matrix. The graphic processor can effectively optimize these operations
The process of constructing features	Requires an accurate determination of the signs and their creation by users	Recognizes high levels based on data and independently creates new signs
Claim to training	The training process is divided into small steps. Then, the results of each step are combined into a single output block	The problem is solved by the method of thorough analysis
Training time	Training takes relatively little time, from a few seconds to several hours	As a rule, the training process takes a long time since the deep learning algorithm includes many levels
Output	The output data is usually a numerical value, for example, assessment or classification	The weekend can have several formats, such as text, estimate or sound

## Data Availability

Most data are contained within the article. All the data are available on request due to restrictions, e.g., privacy or ethics.

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
