# Peer review of "Small Stochastic Data Compactification Concept Justified in the Entropy Basis"

_entropy, 2023, doi:10.3390/e25121567_

Round 1

Reviewer 1 Report

Comments and Suggestions for Authors

This paper is about the long standing problem of reducing the dimension of the representative space used to quantitatively describe features of classes of objects without losing too much information. The motivation of the work is better grasped reading the conclusions, at first.

I try here to rephrase what the authors conclude. 

One gets a quantitative information  on a class of objects by measuring a set of observables (“charcteristic parameters”) on a sample of objects taken from the class of interest. A set of values taken by the chosen observables on one of the objects is an instance. One of the basic problems in general data analysis is to find the optimal number of instances and the optimal (minimal) number of observables, that allow, in the presence of noise, to build regression models, to estimate correlations between observables, to classify and cluster the objects, in a machine learning approach. In this perspective, that is a very relevant one, the authors seem to propose a model of noisy data based on a “Boltzmann entropy” [eq. (6)], that has the form of a conditional, relative entropy. They claim to have found a consistent and tunable method of “compactification” that performs quite well with respect to other established methods such as PCA and random projection methods. 

The paper is intriguing. Perhaps it is a kind of hidden compression sensing (why not to explicitly mention at least some of the literature on “entropy functionals and compressed sensing”?). It is a pity that reading it, managing the bookkeeping of the notations and of the (often missing) definitions is really very heavy. It would require too much time to reconstruct in a clear way the originality of the proposal, and its relative merits. 

I did quite an effort, driven by the curiosity and the relevance of the theme, to get through this work with a positive attitude, but - frankly speaking - I had to give up with the regret of having in my hands a probably relevant piece of work whose validity and impact is too hard to be assessed with a reasonable and affordable reading effort. I believe that in its present form the paper is not acceptable without a radical rewriting. 

As a modest sample of possible hints on how to improve the quality of the manuscript here are a few remarks

compact approximation should be defined (see also line 133)

line 57-58 to uniformly cover a unit [0.1] interval with a frequency 0,01 we need 100 points.

I would use coverage density (define it!) not frequency…not clear

Lines 64-71 Though intuitively sound the consideration made here should be more exactly expressed.

also lines 72-77 is not so clear as it should be

line 90 m and n not defined presumably m number of instances, data and n number of measured observables, descriptors that is the dimension of the data. But, please do specify!

line 94 that is the “individuality” What is that?

line 96 what is the range of the summation?

line 100 what is “this limitation”?

2. Models and methods

line 185 braket not defined (for sure is a scalar product)

line 187 W and its dimensionality  not defined

line 189 censored, not defined

The statements in lines from 189 to 193, crucial to define the model are quite obscure and should be made plainly clear by pedantic control  and definition of every single statement. Do not leave to the reader the burden of making the job for you. 

Example: in line 196 epsilon is a vector of R^n? If this is the case also epsilon + and epsilon - are vectors ….

line 200 what is L(epsilon)?

Line 246 the “Boltzmann” entropy looks like a relative entropy of two distributions (a kind of KL separation?), It is a standard definition?

Line 313 the symbol of scalar product is misprinted.

The captions to the figures are intellectually of no use, they are no more than labels, they do not contribute to the arguments that are poorly referred to in the main text.

Technically, the mathematical analysis of the  model in section 2 seems sensible, perhaps consistent and containing some novelty. But, as said above, it is particularly difficult to assess. The mathematics should be finely presented, without assuming in the reader any prior jargon or acquaintance, like in a classroom of pedestrians.

et cetera.

Comments on the Quality of English Language

see the sample of remarks above. 

Author Response

Dear Colleague,
please read the attached file.
Sincerely, Viacheslav Kovtun.

Reviewer 2 Report

Comments and Suggestions for Authors

The authors explain the shortcomings of traditional machine learning methods in the introduction, but do not explain why the proposed method can solve the problem. The article lacks sufficient and convincing motivations. In the Secition 2.1, the authors do not have any references  in research statement, and the complex description reduces the readability. In the part of proposing the methods, neither definitions nor propositions and theorems are given, and it is difficult to understand the key points of the proposed methods. In terms of experiments, the authors did not compare the method with the methods mentioned in the introduction, and I could not get the conclusion that the proposed method is better. Therefore, I think the author should reconsider the expression of the whole manuscript.

Author Response

(The authors gave the same response as above.)

Reviewer 3 Report

Comments and Suggestions for Authors

I find the paper interesting. However, the biggest drawback is the absence of some more extended experiments. They are needed to check and illustrate the effectiveness of the proposed compactification method in different and reasonable scenarios. 

Minor issues:

1. Lines 93-97. Define variance as sigma^2?

2. In introduction, as a suggestion, it could be interesting for readers to relate compactification with aggregation (magging, neagging, bagging, among others).

3. Lines 352-355. Please correct and clarify.

4. Line 385-393. Please clarify the domain of lambdas and the interpretation.

5. Paper needs a revision for small typos, e.g., lines 193, 218, 290, 297, 313-314, 326, 363, Labels Figures 3 and 4.

Author Response

(The authors gave the same response as above.)

Round 2

Reviewer 1 Report

Comments and Suggestions for Authors

The authors did an effort to meet most of the remarks that were raised. The paper is not perfect in its revised version, but perhaps reached a sufficient level of clarity for interested and willing readers; the theme is relevant, though not yet tersely presented. Perhaps splitting the content into two pieces of work, as also alluded to by the authors in the cover letter, could have been a good idea.

Comments on the Quality of English Language

I have appreciated the effort the authors did to put in a clearer perspective their manuscript. The quality of the English and the overall readability has been improved.

Reviewer 2 Report

Comments and Suggestions for Authors

The authors have addressed all my concerns.

Author Response

(The authors gave the same response as above.)

Reviewer 3 Report

Comments and Suggestions for Authors

The authors improved the paper by adopting some of my suggestions. Some errors remain, but should be detected in typesetting. Regarding the option of not making available any more results from experiments, I believe that at least their existence (with a generic description) could be mentioned in the paper, and they could be made available upon request to the authors.

Author Response

(The authors gave the same response as above.)
